# Pore mutation N617D in the skeletal muscle DHPR blocks Ca$^{2+}$ influx due to atypical high-affinity Ca$^{2+}$ binding

Anamika Dayal[1]*, Monica L Fernández-Quintero[2], Klaus R Liedl[2], Manfred Grabner[1]*

[1]Department of Pharmacology, Medical University of Innsbruck, Innsbruck, Austria; [2]Institute of General, Inorganic and Theoretical Chemistry, University of Innsbruck, Innsbruck, Austria

**Abstract** Skeletal muscle excitation-contraction (EC) coupling roots in Ca$^{2+}$-influx-independent inter-channel signaling between the sarcolemmal dihydropyridine receptor (DHPR) and the ryanodine receptor (RyR1) in the sarcoplasmic reticulum. Although DHPR Ca$^{2+}$ influx is irrelevant for EC coupling, its putative role in other muscle-physiological and developmental pathways was recently examined using two distinct genetically engineered mouse models carrying Ca$^{2+}$ non-conducting DHPRs: DHPR(N617D) (Dayal et al., 2017) and DHPR(E1014K) (Lee et al., 2015). Surprisingly, despite complete block of DHPR Ca$^{2+}$-conductance, histological, biochemical, and physiological results obtained from these two models were contradictory. Here, we characterize the permeability and selectivity properties and henceforth the mechanism of Ca$^{2+}$ non-conductance of DHPR(N617). Our results reveal that only mutant DHPR(N617D) with atypical high-affinity Ca$^{2+}$ pore-binding is tight for physiologically relevant monovalent cations like Na$^+$ and K$^+$. Consequently, we propose a molecular model of cooperativity between two ion selectivity rings formed by negatively charged residues in the DHPR pore region.

*For correspondence:
anamika.dayal@i-med.ac.at (AD);
manfred.grabner@i-med.ac.at (MG)

Competing interests: The authors declare that no competing interests exist.

## Introduction

Excitation-contraction (EC) coupling in skeletal muscle does not require Ca$^{2+}$ influx through the sarcolemmal L-type voltage-gated Ca$^{2+}$ channel Ca$_V$1.1 or dihydropyridine receptor (DHPR), as was convincingly demonstrated in influential studies nearly half a century ago (*Armstrong et al., 1972*; *Schneider and Chandler, 1973*). Contrary to substantial Ca$^{2+}$ influx through cardiac as well as invertebrate muscle DHPRs, which is essential for the Ca$^{2+}$-induced Ca$^{2+}$-release (CICR) mechanism in cardiac-type EC coupling (*Endo, 1977*; *Palade and Györke, 1993*; *Bers, 2002*), Ca$^{2+}$ influx-independent EC coupling in vertebrate skeletal muscle acts by depolarization-induced Ca$^{2+}$ release (DICR). In vertebrate skeletal muscle, voltage-dependent conformational change of the skeletal muscle DHPR is transmitted via protein-protein interaction to the Ca$^{2+}$ release channel - ryanodine receptor (RyR1) in the sarcoplasmic reticulum (SR), inducing its rapid opening. The resulting massive increase in cytosolic Ca$^{2+}$ concentration leads to skeletal muscle contraction (*Armstrong et al., 1972*; *Schneider and Chandler, 1973*; *Rios and Brum, 1987*; *Lamb, 2000*).

Recently, two independently generated genetic mouse models, the EK mouse (*Lee et al., 2015*) and the *nc*DHPR mouse (*Dayal et al., 2017*) revisited the DICR dogma by questioning the role of DHPR Ca$^{2+}$ influx ablation on skeletal muscle performance, fatigue, fiber differentiation, metabolism, and eventually EC coupling. Unexpectedly, despite both the EK and *nc*DHPR mouse models abolish DHPR Ca$^{2+}$ influx, the histological, biochemical, and physiological results obtained from these models are incompatible. The DHPR(E1014K) pore mutation in the EK mouse (*Lee et al., 2015*), besides abolishing Ca$^{2+}$ influx, resulted in reduced SR Ca$^{2+}$ store replenishment during sustained activity,

reduced muscle endurance, decreased muscle protein synthesis, decreased muscle fiber size, a shift in fiber-type specification, and an obese phenotype (*Georgiou et al., 2015*; *Lee et al., 2015*). Conversely, the *nc*DHPR mouse model carrying the DHPR(N617D) pore mutation displayed no differences compared to wild-type (wt) mice across a broad range of tests (*Dayal et al., 2017*). This N→D mutation was previously identified in zebrafish to be responsible for the loss of $Ca^{2+}$ conductance through the DHPR isoform specific for the fast (glycolytic/white) skeletal muscle (*Schredelseker et al., 2010*). Since both the pore mutants, DHPR(E1014K) and DHPR(N617D) preclude $Ca^{2+}$ influx, the striking differences in muscle performance, muscle metabolism, and muscle fiber-type composition between EK and *nc*DHPR mice (*Georgiou et al., 2015*; *Lee et al., 2015*; *Dayal et al., 2017*) are apparently not due to DHPR $Ca^{2+}$ conductance. Instead, the proposed interpretation for the EK mouse was that mutation E1014K alters DHPR selectivity and thus enables permeation of physiologically relevant monovalent cations like $Na^+$ or $K^+$ (*Bannister and Beam, 2011*; *Beqollari et al., 2018*). Nevertheless, permeability and selectivity properties and hence, the mechanism of $Ca^{2+}$ non-conductance of DHPR(N617D) has so far not been investigated thoroughly.

In this study, we demonstrate that the mutant DHPR(N617D) remains $Ca^{2+}$ impermeant even under conditions known to augment L-type $Ca^{2+}$ currents. Our results explicitly show that the DHPR pore mutation N617D leads to an increase in $Ca^{2+}$ pore binding affinity from ~1 μM (characteristic for wt DHPR) to nM range. This more than fourfold enhanced $Ca^{2+}$ binding affinity is sufficient not only to completely block $Ca^{2+}$ conductance through the mutant DHPR(N617D) but also does not allow permeation of monovalent cations like $Cs^+$, $Li^+$, and $Na^+$ under physiological $Ca^{2+}$ concentrations. This pore blocking mechanism due to atypical high-affinity $Ca^{2+}$ binding in mutant DHPR (N617D) strongly contrasts the pore blocking mechanism by low-affinity $Ca^{2+}$ binding in pore mutant DHPR(E1014K). As known from previous studies (*Yang et al., 1993*; *Ellinor et al., 1995*; *Sather and McCleskey, 2003*) any amino acid substitution in the DHPR selectivity filter (EEEE locus) essentially decreases the $Ca^{2+}$ pore binding affinity from μM to mM range, leading to loss of $Ca^{2+}$ selectivity and $Ca^{2+}$ conductance. Based on our recent findings, we propose a molecular model of cooperativity between the divalent cation selectivity (DCS) locus in the outer DHPR pore region (*Cens et al., 2007*) and the EEEE locus in the central pore (*Sather and McCleskey, 2003*). With this model, we can convincingly explain the divergent impacts of both DHPR pore mutations, N617D and E1014K, on $Ca^{2+}$ selectivity and $Ca^{2+}$ conductance and consequently provide an explanation for the incongruences in muscle performance and functioning between the two distinct pore-mutant mouse models. Furthermore, this model of $Ca^{2+}$ selectivity and $Ca^{2+}$ conductance helps us in understanding the $Ca^{2+}$ non-conductance mechanism in previously identified (*Schredelseker et al., 2010*) additional DHPR pore mutations, E→Q and D→K (in the EEEE locus and DCS locus, respectively) that emerged during evolution of other $Ca^{2+}$ non-conducting DHPR isoforms in skeletal muscle of bony fish.

## Results

### DHPR(N617D) is $Ca^{2+}$ impermeant even under current amplifying conditions

To investigate whether DHPR pore mutation N617D obstructs $Ca^{2+}$ permeation also under current enhancing conditions, we implemented corresponding experimental protocols and measured whole-cell $Ca^{2+}$ currents from wt and *nc*DHPR myotubes isolated from new born up to 4-day-old mouse pups. As a first step, inward $Ca^{2+}$ currents were recorded in the presence of 10 μM 1,4-dihydropyridine (DHP) agonist (±)Bay K 8644 applied via the standard bath solution (see Material and methods). For voltage-gated L-type $Ca^{2+}$ channels ($Ca_V$), Bay K 8644 acts as a channel opener by occupying a fenestration site at the interface of repeats III and IV in the pore region (*Grabner et al., 1996*; *Zhao et al., 2019*). Although the standard depolarization protocol (−50 to +80 mV) elicited the expected robust (±)Bay K-induced amplification (p<0.001) of $Ca^{2+}$ currents (No Bay K: $I_{max}$ = −5.04 ± 0.27 pA/pF; $n$ = 9 and with Bay K: $I_{max}$ = −8.82 ± 0.56 pA/pF; $n$ = 6) through the wt DHPR (*Figure 1a*, *center* and *bottom*), no inward $Ca^{2+}$ currents (p<0.001) ($I_{max}$ = −0.02 ± 0.01 pA/pF; $n$ = 5) or tail currents were evoked in *nc*DHPR myotubes under (±)Bay K 8644 administration (*Figure 1a*, *top* and *bottom*).

L-type $Ca^{2+}$ channels show a shift in the mode of gating not only by DHP agonist action (*Hess et al., 1984*) but also in response to strong or prolonged membrane depolarizations. As

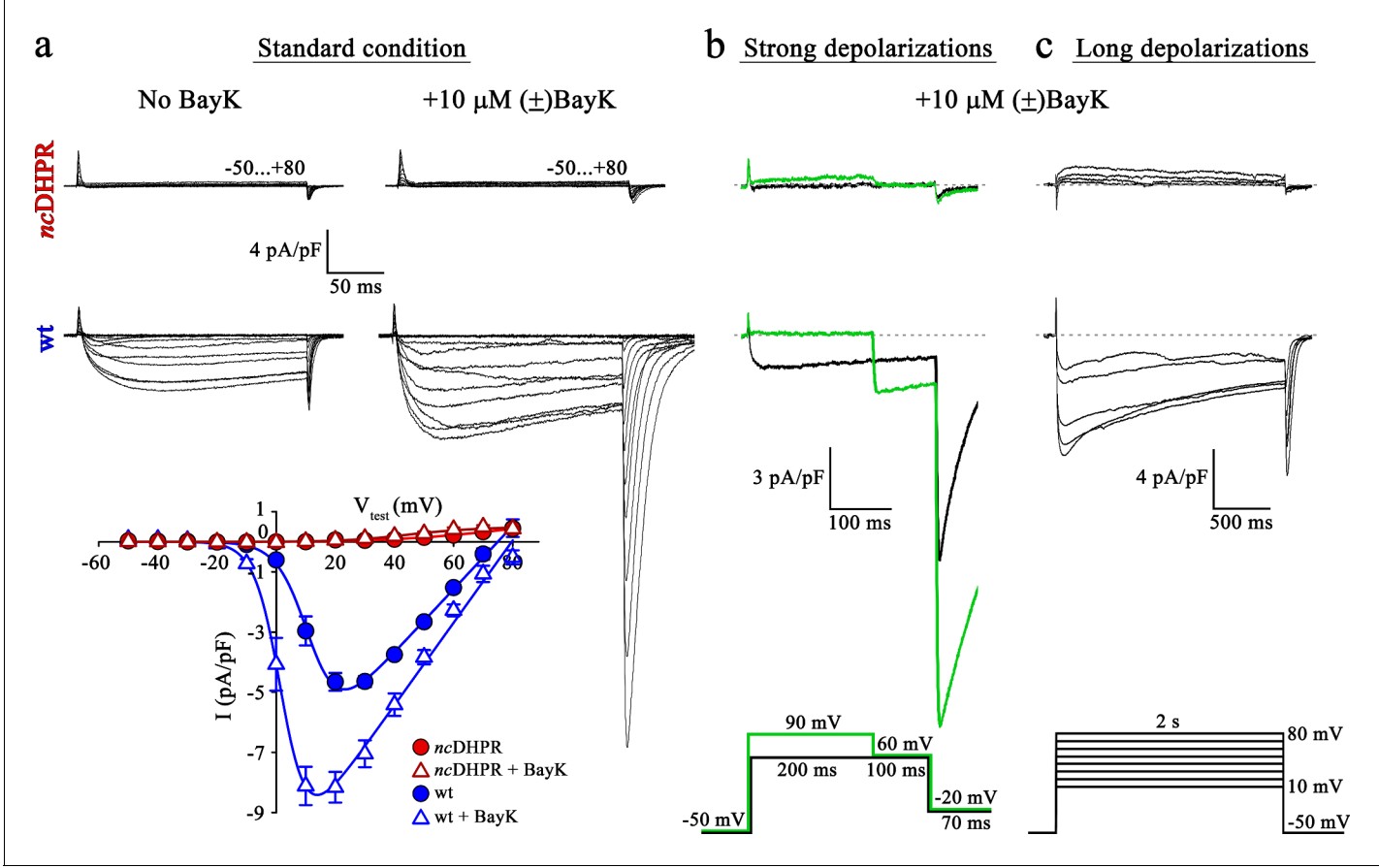

**Figure 1.** Mutant DHPR(N617D) remains $Ca^{2+}$ impermeant despite strong or long depolarizations in the presence of DHP agonist Bay K. (a) Representative whole-cell $Ca^{2+}$ current recordings elicited by 200 ms depolarizations from −50 to +80 mV from *nc*DHPR (*top*) and wt (*center*) myotubes before (*left*) and after (*right*) perfusion with 10 µM of the DHP agonist (±)Bay K 8644. Scale bars, 50 ms (horizontal), 4 pA/pF (vertical). Plots of current-voltage relationship (*bottom*) indicates lack of DHPR inward $Ca^{2+}$ currents in the absence ($I_{max}$ = −0.02 ± 0.01 pA/pF; *n* = 8) and presence ($I_{max}$ = −0.02 ± 0.01 pA/pF; *n* = 5) of (±)Bay K through *nc*DHPR myotubes, in contrast to significant (p<0.001) augmentation of $Ca^{2+}$ currents in wt myotubes upon administration of (±)Bay K (No Bay K: $I_{max}$ = −5.04 ± 0.27 pA/pF; *n* = 9; with Bay K: $I_{max}$ = −8.82 ± 0.56 pA/pF; *n* = 6). (b) 200 ms strong depolarization to +90 mV followed by 100 ms to +60 mV and finally repolarization to −20 mV for 70 ms (*bottom, green lines*) in the presence of 10 µM (±)Bay K, were unable to evoke inward $Ca^{2+}$ currents through DHPR(N617D) (*top*, with +90 mV prepulse: $I_{max}$ = −0.02 ± 0.02 pA/pF; without +90 mV prepulse: $I_{max}$ = 0.01 ± 0.02 pA/pF; *n* = 10). Contrary, wt DHPR displayed significant (p<0.01) depolarization-induced potentiation of inward current at +60 mV (with +90 mV prepulse: $I_{max}$ = −2.97 ± 0.54 pA/pF; without +90 mV prepulse: $I_{max}$ = −1.62 ± 0.37 pA/pF; *n* = 5) (*center*). Upon subsequent repolarization from +60 mV to −20 mV, the tail current was also considerably larger (p<0.01) after the +90 mV pre-conditioning pulse ($I_{tail}$ = −19.36 ± 3.59 pA/pF; *n* = 5) (*center, green trace*) than after the +60 mV pulse ($I_{tail}$ = −10.78 ± 1.99 pA/pF; *n* = 5) (*center, black trace*). Statistical significance was calculated using paired *t*-test. Scale bars, 100 ms (horizontal), 3 pA/pF (vertical). (c) Likewise, 2 s long depolarizations from +10 mV to +80 mV in 10 mV increments (*bottom*) in the presence of 10 µM (±)Bay K, were unable to induce $Ca^{2+}$ influx through DHPR(N617D) (*top*, $I_{max}$ = −0.05 ± 0.02 pA/pF; *n* = 5). The same voltage protocol evoked robust inward $Ca^{2+}$ currents through wt DHPR (*center*, $I_{max}$ = −7.69 ± 0.56 pA/pF; *n* = 5). Scale bars, 500 ms (horizontal), 4 pA/pF (vertical). Data are presented as mean ± SEM; p determined by unpaired Student's *t*-test.
The online version of this article includes the following source data for figure 1:

**Source data 1.** Data for IV graph.

previously demonstrated (*Wilkens et al., 2001*), potentiation of L-type $Ca^{2+}$ channels by DHP agonist Bay K 8644 and strong depolarizations occurs via distinct mechanisms. The shift in mode of gating, also referred to as 'mode 2' gating is characterized at the single-channel level by high open probability ($P_O$) and long mean open times (*Pietrobon and Hess, 1990*). Depolarization-induced entry into mode 2 is reflected by increased $Ca^{2+}$ currents as well as tail currents with slower rate of current decay. To investigate whether strong depolarizations with simultaneous administration of (±) Bay K 8644 enable the entry of mutant DHPR(N617D) into mode 2 and elicit L-type $Ca^{2+}$ currents, we used the pulse protocol depicted in *Figure 1b* (*bottom*) (*Bannister and Beam, 2011*;

*Bannister and Beam, 2013*). Briefly, 200 ms strong, conditioning depolarization pulses from $-50$ mV to +90 mV, followed by a pulse of +60 mV to putatively elicit enhanced inward $Ca^{2+}$ currents and subsequently a repolarization pulse to $-20$ mV to trigger tail currents were applied. As expected from wt myotubes, we recorded significantly larger inward $Ca^{2+}$ current at +60 mV ($I_{max} = -2.97 \pm 0.54$ pA/pF; $n = 5$; p<0.01) as well as tail current at $-20$ mV ($I_{tail} = -19.36 \pm 3.59$ pA/pF; $n = 5$; p<0.01) when preceded by a pulse of +90 mV compared to the corresponding currents recorded without the pre-conditioning depolarization of +90 mV ($I_{max} = -1.62 \pm 0.37$ pA/pF; $I_{tail} = -10.78 \pm 1.99$ pA/pF; $n = 5$) (*Figure 1b*, *center*). Conversely, no inward currents or tail currents could be evoked in *nc*DHPR myotubes with ($I_{max} = -0.02 \pm 0.02$ pA/pF; $n = 10$) or without the +90 mV pre-conditioning pulse ($I_{max} = 0.01 \pm 0.02$ pA/pF; $n = 10$) (*Figure 1b*, *top*). The slight outward component at +90 mV is typically observed at strong depolarizing potentials as described previously (*Schredelseker et al., 2010*; *Dayal et al., 2017*).

Finally, beside strong depolarizations, long depolarizations are known to drive L-type $Ca^{2+}$ channels into mode 2 state (*Pietrobon and Hess, 1990*; *Bannister and Beam, 2013*). Although, the 2 s depolarizations between +10 mV and +80 mV (*Figure 1c*, *bottom*) in the presence of ($\pm$)Bay K 8644 elicited robust, slowly inactivating L-type $Ca^{2+}$ currents in wt control myotubes ($I_{max} = -7.69 \pm 0.56$ pA/pF; $n = 5$) (*Figure 1c*, *center*), no inward $Ca^{2+}$ currents were evoked in *nc*DHPR myotubes ($I_{max} = -0.05 \pm 0.02$ pA/pF; $n = 5$) (*Figure 1c*, *top*), suggesting that DHPR(N617D) remained $Ca^{2+}$ impermeant even under potentiating conditions. We found slight outward currents that were similar to previously observed currents in DHPR$\alpha_{1S}$-null (*dysgenic*) myotubes (*Bannister and Beam, 2013*) recorded under identical conditions, and thus are unrelated to the DHPR.

Altogether, our results demonstrate that recording conditions known to potentiate L-type inward $Ca^{2+}$ currents through the wt DHPR were unable to evoke $Ca^{2+}$ currents through the mutant DHPR (N617D) in the *nc*DHPR mouse model. Out of the three, so far described mutant mammalian DHPR $Ca^{2+}$ channels with ablated $Ca^{2+}$ conducting ability under standard recording conditions, namely R174W (*Eltit et al., 2012*), E1014K (*Lee et al., 2015*), and N617D (*Dayal et al., 2017*), only the voltage-sensor mutant R174W opened partially and produced tail currents under ($\pm$)Bay K 8644 administration. This malignant hyperthermia-linked DHPR voltage-sensor mutant R174W also displayed small, but clearly detectable inward $Ca^{2+}$ currents together with enhanced tail currents in response to strong or prolonged depolarizations in the presence of ($\pm$)Bay K 8644 (*Bannister and Beam, 2013*). Integrating previous and present results (*Bannister and Beam, 2011*), we can conclude that it is impossible to force either of the two DHPR pore mutants, DHPR(N617D) and DHPR(E1014K) into a $Ca^{2+}$ conducting mode by executing the above-described L-type $Ca^{2+}$ current amplifying conditions.

## DHPR(N617D) does not conduct Na$^+$ currents

Since both pore mutants, DHPR(N617D) as well as DHPR(E1014K) strictly prevent $Ca^{2+}$ influx even under current enhancing conditions, the striking differences in muscle performance, metabolism, and fiber-type composition between *nc*DHPR and EK mice (*Lee et al., 2015*; *Georgiou et al., 2015*; *Dayal et al., 2017*) can evidently not be due to DHPR $Ca^{2+}$ conductance. However, the reason for these puzzling phenotypic differences could be attributed to distinct selectivity and permeation properties of physiologically relevant monovalent cations through these mutated DHPRs. Basic biophysical characterization of both the DHPR pore mutants, performed either in the respective mouse model (*Lee et al., 2015*; *Dayal et al., 2017*) or in heterologous expression systems (*Dirksen and Beam, 1999*; *Schredelseker et al., 2010*; *Bannister and Beam, 2011*; *Beqollari et al., 2018*) already pointed out substantial differences in monovalent cation conductance. Specifically, under standard $Ca^{2+}$ current recording conditions with 145 mM $Cs^+$ present in the patch pipette to block $K^+$ channels (*Clay and Shlesinger, 1984*), massive outward $Cs^+$ currents through DHPR(E1014K) (*Bannister and Beam, 2011*; *Lee et al., 2015*; *Beqollari et al., 2018*) but not through DHPR (N617D) (*Schredelseker et al., 2010*; *Dayal et al., 2017*; *Beqollari et al., 2018*) were observed (see also *Figure 1a*, *top* and *bottom* and *Figure 2b*, *bottom*).

Apparently, the question arose if this $Cs^+$ leakiness of DHPR(E1014K) and tightness of DHPR (N617D) is also factual for other monovalent cations like the physiologically relevant Na$^+$ ions. To clarify this conundrum, we performed patch-clamp recordings on *nc*DHPR myotubes under comparable experimental conditions like previously used on DHPR(E1014K) expressed in *dysgenic* myotubes (*Bannister and Beam, 2011*).

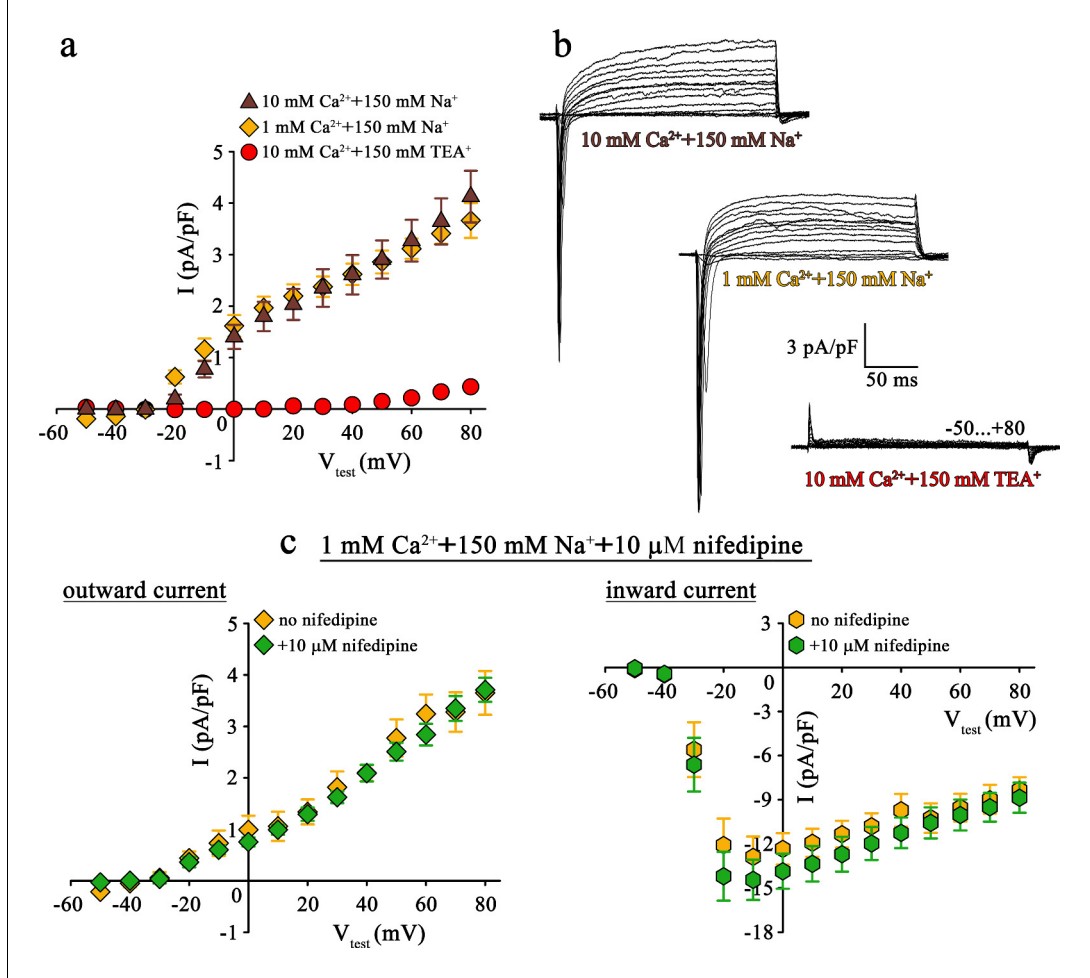

**Figure 2.** Mutant DHPR(N617D) does not conduct inward $Na^+$ currents in the presence of near physiological [$Na^+$]. (a) Plots of current-voltage relationship for DHPR-mediated $Na^+$ currents recorded from *nc*DHPR myotubes indicate the absence of slow-activating, non-inactivating inward $Na^+$ currents in the presence of near physiological 150 mM external $Na^+$ with either 10 mM ($n = 8$) or 1 mM external $Ca^{2+}$ ($n = 9$). Control recordings were performed in standard bath solution (150 mM $TEA^+$, 10 mM $Ca^{2+}$) ($n = 8$). (b) Representative current recordings from *nc*DHPR myotubes in response to 200 ms depolarizing test pulses between −50 to +80 mV in the presence of 10 mM $Ca^{2+}$ with either 150 mM $Na^+$ (*top*) or 150 mM $TEA^+$ (*bottom*), or 1 mM $Ca^{2+}$ with 150 mM $Na^+$ (*center*) in the bath solution. Scale bars, 50 ms (horizontal), 3 pA/pF (vertical). (c) Plots of current-voltage relationship for *nc*DHPR myotubes at 150 mM external $Na^+$ and 1 mM external $Ca^{2+}$ indicate no difference (p>0.05) in outward and inward currents in the presence ($n = 10$) and absence ($n = 6$) of 10 μM of the 1,4-DHP antagonist nifedipine.

The online version of this article includes the following source data for figure 2:

**Source data 1.** Data for IV graphs.

As demonstrated in *Figure 2a and b* (*top*), at near physiological (150 mM) external $Na^+$ as the sole monovalent cation and 10 mM external $Ca^{2+}$, no slow inward $Na^+$ currents resembling the L-type $Ca^{2+}$ currents were observed in *nc*DHPR myotubes ($n = 8$). Even upon reducing external $Ca^{2+}$ from 10 to 1 mM to lower the blocking effect by $Ca^{2+}$, DHPR(N617D) remained fully impermeant to $Na^+$ ions (*Figure 2a and b*, *center*; $n = 9$). Both these observations were in stark contrast to the robust slow-activating, non-inactivating inward $Na^+$ currents recorded from DHPR(E1014K) under comparable conditions (*Bannister and Beam, 2011*). The large, rapidly activating and inactivating inward currents observed within the first ~20 ms after the onset of test potentials can be ascribed to endogenous skeletal muscle $Na^+$ channel ($Na_V$) isoforms (*Numann et al., 1994*) and were to a big extent, present even after administration of 2 μM $Na^+$ channel blocker tetrodotoxin (*Bannister and Beam, 2011*). The outward currents observed in the presence of 150 mM external $Na^+$ (*Figure 2a and b*, *top* and *center*) can certainly be ruled out to be $Cs^+$ currents through DHPR(N617D) since they are blocked by the $K^+$ channel blocker $TEA^+$ (*Figure 2b*, *bottom*; $n = 8$) and show kinetics very

different from $Cs^+$ currents recorded from DHPR(E1014K) (*Lee et al., 2015*; *Bannister and Beam, 2011*; *Beqollari et al., 2018*). Moreover, the current-voltage relationship of these currents which start from approximately −30 mV is linear (10 mM $Ca^{2+}$: $R^2$ = 0.99; 1 mM $Ca^{2+}$: $R^2$ = 0.97) and this together with $TEA^+$ sensitivity strongly points to residual $K^+$ currents through the endogenous delayed rectifier $K^+$ channel ($K_V$) (*DiFranco and Vergara, 2011*). Despite >10 min perfusion with 145 mM $Cs^+$ from the patch pipette, these putative $K_V$ currents remained unblocked, probably due to limited diffusion in fairly elongated and narrow myotubes.

To directly test for a putative contribution of DHPR(N617D) in mediating the outward and inward currents described above (*Figure 2a and b*), we measured whole-cell currents in the presence of the 1,4-DHP $Ca^{2+}$ antagonist nifedipine. As depicted in *Figure 2c*, patch-clamp recordings performed upon addition of 10 μM nifedipine to the bath solution containing 1 mM $Ca^{2+}$ and 150 mM $Na^+$, exhibited nifedipine-insensitive slow outward currents (*Figure 2c*, *left*) and rapidly activating and inactivating inward currents (*Figure 2c*, *right*). Current-voltage relationship of outward currents (no nifedipine: $R^2$ = 0.98; with nifedipine: $R^2$ = 0.98) as well as of inward currents (no nifedipine: $I_{max}$ = −13.22 ± 1.41 pA/pF; n = 6; with nifedipine: $I_{max}$ = −14.96 ± 1.40 pA/pF; n = 10) were unaffected (p>0.05) by the presence of nifedipine. These results unambiguously confirm that DHPR(N617D) is not accountable for the outward and inward currents observed in the presence of near physiological external $Na^+$.

## Aberrant high-affinity $Ca^{2+}$ binding to the DHPR(N617D) channel pore

As pointed out above, our results together with previous work (*Dirksen and Beam, 1999*; *Schredelseker et al., 2010*; *Bannister and Beam, 2011*; *Lee et al., 2015*; *Dayal et al., 2017*; *Beqollari et al., 2018*) clearly demonstrate substantial distinct pore properties between DHPR (E1014K) and DHPR(N617D). Although both DHPR pore mutants do not conduct $Ca^{2+}$, DHPR (E1014K) additionally lost its ion-selectivity and robustly conducts monovalent anions like $Cs^+$ as well as physiologically relevant $Na^+$ and $K^+$ even in the presence of physiological concentrations of external $Ca^{2+}$. Although the channel properties of DHPR(E1014K), with its charge conversion of selectivity filter glutamate $E_{1014}$, are accurately explained by a widely accepted model of cardiac $Ca^{2+}$ channel selectivity and permeation (*Yang et al., 1993*; *Ellinor et al., 1995*; *Sather and McCleskey, 2003*) (see Discussion), the non-conductance mechanism of DHPR(N617D) is still unknown (*Schredelseker et al., 2010*; *Dayal et al., 2017*).

Consequently, we wanted to test if the additional negative charge introduced via the N617D substitution, three residues C-terminal to the selectivity filter glutamate in repeat II and positioned towards the pore entrance (*Dayal et al., 2017*), enhances $Ca^{2+}$ affinity to the pore and resultantly blocks functional $Ca^{2+}$ permeation by hampering the electrostatic repulsion mechanism (*Sather and McCleskey, 2003*). As a direct index of $Ca^{2+}$ pore-binding affinity assessment, we performed $Ca^{2+}$ block of $Li^+$ current experiments on *nc*DHPR and wt myotubes. In the presence of 100 mM extracellular $Li^+$ and without extracellular $Ca^{2+}$ block (free $[Ca^{2+}]=0$), inward $Li^+$ currents were indistinguishable (p>0.05) between *nc*DHPR ($I_{max}$ = −2.32 ± 0.35 pA/pF; n = 16) and wt ($I_{max}$ = −2.07 ± 0.47 pA/pF; n = 9) myotubes (*Figure 3a*). However, increase in extracellular $Ca^{2+}$ concentration to 1 μM showed a highly significant (p<0.001) reduction of $Li^+$ currents through *nc*DHPR ($I_{max}$ = −0.47 ± 0.10 pA/pF; n = 8) compared to wt ($I_{max}$ = −1.68 ± 0.25 pA/pF; n = 6) myotubes (*Figure 3b*). These results indicate a higher efficiency of $Ca^{2+}$ block of $Li^+$ currents due to enhanced $Ca^{2+}$ binding affinity to the DHPR(N617D) pore. In particular, at 3 μM external $Ca^{2+}$, no $Li^+$ currents could be evoked from *nc*DHPR myotubes ($I_{max}$ = −0.06 ± 0.02 pA/pF; n = 7) but small, significant (p<0.001) currents through the wt DHPR ($I_{max}$ = −0.24 ± 0.04 pA/pF; n = 6) were still existent (*Figure 3c*). A complete list of peak inward $Li^+$ currents at varying free external $Ca^{2+}$ concentrations is presented in *Table 1*.

To directly validate if the slow inward $Li^+$ currents are conducted by DHPR(N617D), we recorded $Li^+$ currents in *nc*DHPR myotubes in the presence of the 1,4-DHP antagonist nifedipine. As depicted in *Figure 4*, recordings performed upon addition of 10 μM nifedipine to the bath solution containing 0 $Ca^{2+}$ exhibited a drastic reduction (p<0.001) of slow inward $Li^+$ currents (no nifedipine: $I_{max}$ = −2.41 ± 0.27 pA/pF; n = 11; with nifedipine: $I_{max}$ = −0.35 ± 0.13 pA/pF; n = 16). These results confirm that the slow inward $Li^+$ currents observed in the absence of external $Ca^{2+}$ are mediated by DHPR(N617D).

Large, rapidly activating and inactivating inward currents detected in both wt and *nc*DHPR myotubes within the first ~20 ms of the onset of test potentials are $Li^+$ currents through endogenous

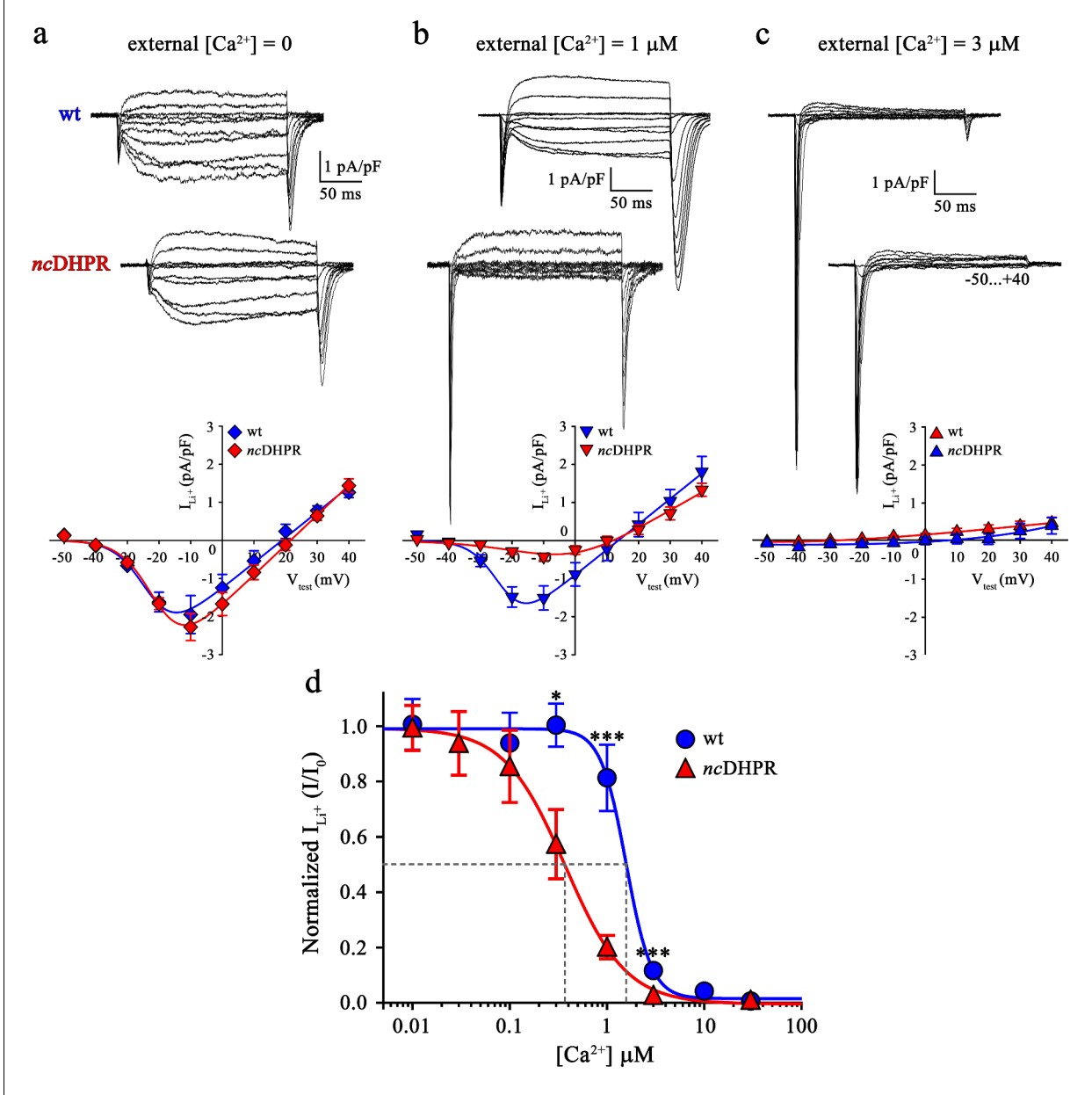

**Figure 3.** Binding of $Ca^{2+}$ ions with nanomolar affinity within the pore of mutant DHPR(N617D) precludes $Ca^{2+}$ permeation. Representative whole-cell $Li^+$ current recordings from wt and *nc*DHPR myotubes in response to 200 ms depolarizations from −50 to +40 mV in the presence of 100 mM external $Li^+$ and either 0 (**a**), 1 μM (**b**) or 3 μM (**c**) free external $Ca^{2+}$. Scale bars, 50 ms (horizontal), 1 pA/pF (vertical). Plots of current-voltage relationship are depicted at the bottom of the corresponding representative $Li^+$ current traces. Inward $Li^+$ currents with no blocking ion (free $[Ca^{2+}]=0$) were indistinguishable (p>0.05) between *nc*DHPR ($I_{max}$ = −2.32 ± 0.35 pA/pF; n = 16) and wt ($I_{max}$ = −2.07 ± 0.47 pA/pF; n = 9) myotubes (**a**, *bottom*). However, at higher external $[Ca^{2+}]$ of 1 μM (**b**) and 3 μM (**c**), inward $Li^+$ currents were significantly (p<0.001) smaller in *nc*DHPR ($I_{max}$ = −0.47 ± 0.10 pA/pF, n = 8; $I_{max}$ = −0.06 ± 0.02 pA/pF; n = 7, respectively) compared to wt myotubes ($I_{max}$ = −1.68 ± 0.25 pA/pF, n = 6; $I_{max}$ = −0.24 ± 0.04 pA/pF; n = 6, respectively). (**d**) Four-parameter fitted concentration-response curves of $Ca^{2+}$ block of inward $Li^+$ currents for wt and mutant *nc*DHPR. Averaged $I/I_0$ peak currents are plotted as a function of free external $Ca^{2+}$ concentrations (up to 30 μM) and each data point is an average of 5–16 myotubes (*Table 1*). There is a significant (p<0.01) shift in $IC_{50}$ (*grey dotted lines*) between wt ($IC_{50}$ = 1.57 μM) and *nc*DHPR ($IC_{50}$ = 0.37 μM) indicating a 4.2-fold higher $Ca^{2+}$ pore-binding affinity in the mutant DHPR(N617D) channel. Data are presented as mean ± SEM; p determined by unpaired Student's *t*-test. The online version of this article includes the following source data for figure 3:

**Source data 1.** Data for dose-response graph.

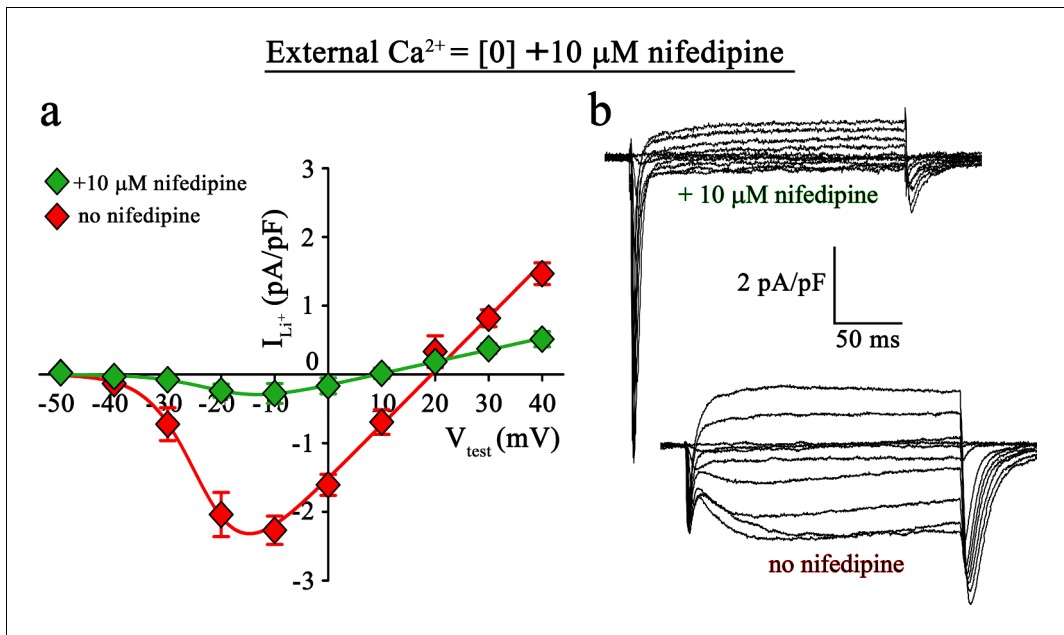

**Figure 4.** Inward Li$^+$ currents conducted by DHPR(N617D) are sensitive to nifedipine block. (**a**) Plots of current-voltage relationship for DHPR-mediated Li$^+$ currents recorded from *nc*DHPR myotubes in the presence (I$_{max}$ = −0.35 ± 0.13 pA/pF; *n* = 16) and absence (I$_{max}$ = −2.41 ± 0.27 pA/pF; *n* = 11) of 10 µM of the 1,4-DHP antagonist nifedipine, 100 mM external Li$^+$, and free external Ca$^{2+}$ = [0]. Maximum inward Li$^+$ currents were significantly (p<0.001) reduced in the presence of nifedipine. (**b**) Representative whole-cell Li$^+$ current recordings from *nc*DHPR myotubes in response to 200 ms depolarizations from −50 to +40 mV in the presence (*upper*) and absence (*lower*) of 10 µM nifedipine with 100 mM external Li$^+$ and 0 external Ca$^{2+}$ concentration. Scale bars, 50 ms (horizontal), 2 pA/pF (vertical). Data are presented as mean ± SEM; p determined by unpaired Student's *t*-test. The online version of this article includes the following source data for figure 4:

**Source data 1.** Data for IV graph.

---

skeletal muscle Na$^+$ channels, Na$_V$ (**Numann et al., 1994**; **DiFranco and Vergara, 2011**). Interestingly, their amplitudes appear to correlate negatively to the slow Li$^+$ current amplitudes through wt or mutant N617D DHPRs at different external Ca$^{2+}$ concentrations (**Figure 3a-c**) and were similarly amplified upon nifedipine block of DHPR(N617D) channels at external free [Ca$^{2+}$]=0 (**Figure 4b**). A

---

**Table 1.** Effect of varying free external Ca$^{2+}$ concentrations on peak inward Li$^+$ currents (I$_{max}$) in wt and *nc*DHPR myotubes.

I$_{max}$ values of inward I$_{Li+}$ are represented as mean ± SEM with corresponding number of recordings (*n*) from wt and *nc*DHPR myotubes. *p<0.05; ***p<0.001, unpaired Student's *t*-test.

| Free [Ca$^{2+}$] | wt | | *nc*DHPR | |
| --- | --- | --- | --- | --- |
| | I$_{max}$ (pA/pF) | *n* | I$_{max}$ (pA/pF) | *n* |
| 0 | −2.07 ± 0.47 | 9 | −2.32 ± 0.35 | 16 |
| 10 nM | −2.08 ± 0.19 | 6 | −2.30 ± 0.19 | 12 |
| 30 nM | – | – | −2.17 ± 0.27 | 12 |
| 100 nM | −1.94 ± 0.23 | 5 | −1.98 ± 0.30 | 9 |
| 300 nM | −2.08 ± 0.16 | 8 | −1.33 ± 0.29 * | 8 |
| 1 µM | −1.68 ± 0.25 | 6 | −0.47 ± 0.10 *** | 8 |
| 3 µM | −0.24 ± 0.04 | 6 | −0.06 ± 0.02 *** | 7 |
| 10 µM | −0.09 ± 0.05 | 6 | – | – |
| 30 µM | −0.01 ± 0.02 | 5 | −0.02 ± 0.02 | 8 |

possible competition between $Ca_V$ and $Na_V$ channels for $Li^+$ ions, with $Ca_V$ taking the priority was not investigated further in the present study.

Eventually, to quantify the impact of the DHPR pore mutation N617D on $Ca^{2+}$ pore binding-affinity in comparison to wt DHPR, $Ca^{2+}$ concentration-response curves displaying inhibition of peak inward $Li^+$ currents by $Ca^{2+}$ were analyzed by a nonlinear fit with variable slope (four parameter). As demonstrated in *Figure 3d*, $Ca^{2+}$ block of inward $Li^+$ currents through the skeletal muscle wt DHPR displays an $IC_{50}$ of 1.57 µM (95% CI: 1.36–1.80 µM), which is highly comparable to published values of cardiac DHPR $Ca^{2+}$ pore binding affinity (*Yang et al., 1993*; *Ellinor et al., 1995*; *Cibulsky and Sather, 2000*; *Sather and McCleskey, 2003*). Interestingly, pore mutant DHPR(N617D) exhibited an $IC_{50}$ of 372.8 nM (95% CI: 334.4–415.8 nM) which is indeed 4.2-fold shifted to lower $Ca^{2+}$ concentrations. Thus, in the mutant DHPR(N617D), introduction of the negatively charged residue $D_{617}$ into the DHPR pore in close vicinity of the selectivity filter EEEE results in a significant (p<0.01) decrease in $IC_{50}$ from µM to nM concentrations. Notably, also the Hill coefficient ($n_H$) was significantly (p<0.01) different between wt DHPR (−3.32; 95% CI: −4.68 – −2.61) and DHPR(N617D) (−1.39; 95% CI: −1.59 – −1.23) (see Discussion). Since atypical high-affinity binding of $Ca^{2+}$ to the mutant pore is apparently incompatible with $Ca^{2+}$ conductance, this supports the idea of a mechanism by which the mutant DHPR(N617D) pore is occluded.

## Discussion

Our results and earlier findings of *Bannister and Beam, 2011* show that in contrast to DHPR (N617D), DHPR(E1014K) functions as a slow-activating, non-inactivating, junctionally-targeted inward $Na^+$ channel. Indeed, this difference in intramuscular $Na^+$ conductance could be one of the reasons for the different phenotypes observed with the two non-$Ca^{2+}$ conducting DHPR pore-mutant mouse strains, *nc*DHPR and EK. However, of higher physiological relevance than this $Na^+$ conductance of DHPR(E1014K), is probably its additional massive 1,4-DHP-sensitive, non-inactivating outward $K^+$ conductance, which again is completely absent in the DHPR(N617D) counterpart (*Beqollari et al., 2018*). $K^+$ accumulation is known to play a crucial role in muscle fatigue (*Allen et al., 2008*) and hence, in the EK mouse strain this mutationally introduced $K^+$ efflux from cytoplasm into the transvers (t)-tubular lumen may exacerbate muscle fatigability during periods of enhanced, repetitive activity (*Beqollari et al., 2018*). Additionally, $K^+$ overload in the t-tubule is expected to induce aberrant muscle membrane excitability, which might be the root cause for the muscle histological and metabolic aberrations observed in EK (*Georgiou et al., 2015*; *Lee et al., 2015*) but not in *nc*DHPR mice (*Dayal et al., 2017*).

Even though the results and interpretations of *Beqollari et al., 2018* intriguingly suggest $K^+$ permeability through DHPR(E1014K) as the basis for the biophysical differences between mutant strains *nc*DHPR and EK, we have to take into consideration the fact that these data were derived from heterologous expression studies in tsA-201cells and not from isolated EK skeletal muscle fibers. Moreover, considering the short duration (~5 ms) of the skeletal muscle action potential (AP) (*Sperelakis et al., 2012*), slow DHPR activation kinetics (*Schrötter et al., 2017*), and relatively strong depolarization (+20 mV) required for detectable $K^+$ currents (*Beqollari et al., 2018*), further studies on intact EK fibers are required to fully understand if $K^+$ flux through DHPR(E1014K) during an AP or series of APs has important phenotypic implications or not.

Besides the putative influence of the $Na^+$ and $K^+$ leakiness of DHPR(E1014K) on the EK phenotype, *Lee et al., 2015* and *Georgiou et al., 2015* presented an alternative hypothesis, which could particularly explain the metabolic aberrations in the EK mouse model. It was proposed that $Ca^{2+}$ permeation and/or high-affinity $Ca^{2+}$ binding to the DHPR is conformationally coupled to the activation of $Ca^{2+}$ / calmodulin-dependent protein kinase type II (CaMKII) and SR store refilling during sustained muscle activity. Consequently, lack of high-affinity $Ca^{2+}$ binding to the DHPR(E1014K) pore causes a decrease in these $Ca^{2+}$-dependent enzyme activities, ensuing alterations in the downstream Ras/Erk/mTORC1 signaling pathways and as a result decreased muscle protein synthesis and the described muscle physiological aberrations (*Georgiou et al., 2015*; *Lee et al., 2015*). Although the results derived from *nc*DHPR mice (*Dayal et al., 2017*) exclude the significance of DHPR $Ca^{2+}$ permeation, they are consistent with a putative crucial role of high-affinity DHPR $Ca^{2+}$ pore binding (like found in wt DHPR or DHPR(N617D)) for accurate CaMKII activation and thus, intact downstream signaling.

Integration of our recent and previous findings (*Bannister and Beam, 2011*; *Beqollari et al., 2018*) helped us in addressing the following questions: How to understand the obvious distinct origin of the non-conductance mechanisms of mutants DHPR(E1014K) and DHPR(N617D)? Why is DHPR(E1014K) leaky for monovalent cations, but DHPR(N617D) preserves its high selectivity for $Ca^{2+}$ ions?

## DHPR pore residues responsible for $Ca^{2+}$ selectivity and $Ca^{2+}$ permeation

In an attempt to answer the above questions, we intend to expand a widely accepted molecular model of $Ca^{2+}$ channel selectivity and permeation based on two elegant studies from the Tsien lab (*Yang et al., 1993*; *Ellinor et al., 1995*), and comprehensively discoursed in the review of *Sather and McCleskey, 2003*. According to this model, one $Ca^{2+}$ ion binds to a single high-affinity site formed by all four glutamates (EEEE locus) of the DHPR selectivity filter. This tight embracement of $Ca^{2+}$ in the DHPR pore is a prerequisite for the high selectivity for $Ca^{2+}$ over $Na^+$, $K^+$, or other monovalent cations. However, to enable rapid passage of $Ca^{2+}$ through the pore, a two-site mechanism that overcomes this tight $Ca^{2+}$ binding is essential. Accordingly, the EEEE locus has been suggested to be physically flexible. Hence, irrespective that all four selectivity filter glutamates are needed to hold a single $Ca^{2+}$ ion with high affinity ($K_D$ ~1 µM), their conformation can rapidly rearrange to accommodate a pair of $Ca^{2+}$ ions within the pore, but then both bound with much lower affinity (apparent $K_D$ ~14 mM). This intermediate short-lived low-affinity state, together with a $Ca^{2+}$-$Ca^{2+}$ repulsion mechanism occurring in this doubly occupied pore, whereby one of the occupying $Ca^{2+}$ ions is pushed out to the cytosolic side, is the basis for fast $Ca^{2+}$ ion passage through the pore.

Although the $Ca^{2+}$ selectivity filter in form of the conserved EEEE locus within the pore of high threshold voltage-gated $Ca^{2+}$ channels (HVA VGCC) satisfactorily explains divalent/monovalent ion selection, it neither explains the differences in the selectivity for $Ca^{2+}$ among other divalent ions nor the observed distinct conductances through the different HVA VGCC isoforms (*Cens et al., 2007*). In their interesting study, *Cens et al., 2007* via point mutational analyses and molecular modeling identified a ring of non-conserved negatively charged residues located at homologous positions in each of the four repeats of the DHPR pore, which were responsible for the distinct channel profiles. This ring coined as 'divalent cation selectivity' (DCS) locus, is present in different constellations in every VGCC and is located towards the outer channel pore region in close vicinity of the selectivity filter EEEE locus. The DCS locus might constitute an additional, low-affinity $Ca^{2+}$-binding site which, together with distinct negative charges closely adjacent to the EEEE locus (*Williamson and Sather, 1999*), plays a crucial role in defining and directly participating in the generation of different $Ca^{2+}$ conductances in different HVA $Ca^{2+}$ channels (*Cens et al., 2007*).

## $Ca^{2+}$ non-selectivity and $Ca^{2+}$ non-permeability of the mutant DHPR (E1014K)

As discussed above, proper $Ca^{2+}$ channel permeation and high selectivity are essentially dependent on a single high-affinity $Ca^{2+}$-binding site formed by all four glutamates of the DHPR selectivity filter to assure tight embracement of $Ca^{2+}$. Any substitution in the EEEE locus abolishes/decreases this high (µM) $Ca^{2+}$ pore binding affinity as demonstrated by $Ca^{2+}$ block of $Li^+$ current experiments (*Yang et al., 1993*; *Ellinor et al., 1995*; *Sather and McCleskey, 2003*). Specifically, the strongest impact on the binding affinity was produced by exchange of E in repeat III. The EIIIK mutation drastically reduced the pore's affinity for $Ca^{2+}$ to 1000-fold, as is depicted by an increase in $IC_{50}$ from ~1 µM to ~1 mM for $Ca^{2+}$ block of $I_{Li+}$ (*Yang et al., 1993*). Although these classical affinity experiments where performed in the cardiac DHPR, the comprehended selectivity/conductance model appears to be congruent with the skeletal muscle DHPR. Accordingly, the large outward $Cs^+$ current found in the skeletal muscle EIIIK mutant DHPR(E1014K) (*Bannister and Beam, 2011*; *Lee et al., 2015*; *Beqollari et al., 2018*), which was not blocked even in the presence of 10 mM external $Ca^{2+}$, was consequently interpreted as an indication of very little residual $Ca^{2+}$ binding within the DHPR (E1014K) pore (*Dirksen and Beam, 1999*; *Beqollari et al., 2018*). Similarly, a considerable inward $Na^+$ current through EK myotubes despite external $Ca^{2+}$ concentration as high as 10 mM (*Bannister and Beam, 2011*) again indicates a very marginal, low-affinity binding of $Ca^{2+}$ within the DHPR(E1014K) pore. Consequently, low-affinity pore-bound $Ca^{2+}$ is unable to block the flux of any

cation in both directions and hence $Ca^{2+}$ selectivity is abolished in the mutant DHPR(E1014K). In addition, since the EEEE locus is mutated to EEKE, attraction of a second $Ca^{2+}$ and subsequent competition for binding valences with the $Ca^{2+}$ ion that is already bound with low affinity to this EEKE locus is impossible. Absence of this intermediate doubly occupied pore and thus, of the $Ca^{2+}$- $Ca^{2+}$ repulsion mechanism as the basis for fast, unidirectional $Ca^{2+}$ ion passage through the pore is sufficient to explain the lack of $Ca^{2+}$ conductance through the mutant DHPR(E1014K).

## High $Ca^{2+}$ selectivity and $Ca^{2+}$ non-permeability of the mutant DHPR (N617D)

Now the question arose, how to understand the pore blocking mechanism observed in DHPR (N617D) by coalescing the models discussed above? *Figure 5a* depicts the putative mechanism of $Ca^{2+}$ conductance through wt DHPR. The carboxyl oxygens of the DCS locus point toward the pore lumen, allowing coordination of incoming divalent cations with a preference for $Ca^{2+}$ (*Cens et al., 2007*). According to our postulated pore model (*Figure 5a*), $Ca^{2+}$ ions from the t-tubular (extracellular) side are attracted to the negative charges of the DCS locus, which in mouse $DHPR\alpha_{1S}$ is formed by $D_{296}$ of repeat I, $E_{1327}$ of repeat IV, and supported by $D_{615}$ of repeat II. This loosely bound $Ca^{2+}$ ion is easily mobilized (probably by charge repulsion from excess $Ca^{2+}$ ions in the t-tubule) and migrates deeper into the pore to compete with the tightly bound $Ca^{2+}$ ion for binding valences of the EEEE locus (in mouse skeletal-muscle DHPR: $E_{292}$, $E_{614}$, $E_{1014}$, $E_{1323}$). Henceforth, due to the reduced binding (µM to mM affinity), $Ca^{2+}$- $Ca^{2+}$ repulsion (*Sather and McCleskey, 2003*) takes place, eventually pushing the loosely bound $Ca^{2+}$ into the cytosol. This conceptual model is supported by simulation experiments as depicted in *Figure 6*. Molecular dynamics simulations show that the EEEE locus attracts and stabilizes a single $Ca^{2+}$ ion (*Figure 6b–c*). However, in the wt DHPR we also observe conformational changes in the EEEE locus that allow binding of a second $Ca^{2+}$ ion. This additional $Ca^{2+}$ ion results in a weaker binding of the glutamate residues to both $Ca^{2+}$ ions, thereby causing a repulsion between the two ions, which is reflected in their decreasing distance to as low as 6 Å (*Figure 6c*, *left*). Furthermore, metadynamics simulations show that as a consequence of this $Ca^{2+}$ - $Ca^{2+}$ repulsion occurring in the doubly occupied EEEE locus, one of the two $Ca^{2+}$ ions moves toward the cytosolic side (*Figure 6c*, *left*; *Figure 6—video 1*). The weaker binding of the $Ca^{2+}$ ions to the EEEE locus of the wt DHPR compared to the mutant DHPR(N617D), is reflected in the significantly (p<0.001) lower free energy barrier (*Figure 6d*).

Contrary to this smooth $Ca^{2+}$- conducting mechanism of wt DHPR, the additional negative charge $D_{617}$ in mutant DHPR(N617D), introduced in the close vicinity to the residue $D_{615}$ in repeat II (*Figure 5b*), creates an additional binding valence and as a result induces an aberrant high $Ca^{2+}$ binding-affinity to the DCS locus. According to our model, this considerably tighter bound $Ca^{2+}$ is consequently not sufficiently mobile anymore to travel deeper into the pore to compete for the binding valences of the selectivity-filter EEEE locus with the already strongly bound $Ca^{2+}$ ion. Overall, lack of formation of the intermediate short-lived lower-affinity $Ca^{2+}$ binding state, together with the consequential lack of $Ca^{2+}$- $Ca^{2+}$ repulsion at the EEEE locus explicitly explains the absence of $Ca^{2+}$ influx through the DHPR(N617D) pore. Congruently, molecular dynamics simulations show that immediately after the equilibration step, one $Ca^{2+}$ ion is stabilized at the EEEE locus while the other $Ca^{2+}$ is bound to the DCS locus (*Figure 6c*, *right*; *Figure 6—video 2*). This translocation of the $Ca^{2+}$ ions to the DCS and EEEE locus occurs already within 1 ns of simulation time succeeding the last step of the equilibration protocol. Here, the distance between the two $Ca^{2+}$ ions is ~9 Å. The strong binding of the two $Ca^{2+}$ ions to the EEEE and DCS locus makes it impossible for any other ion, like $Li^+$, to pass through the DHPR(N617D) pore. Thus, simulations of pulling of $Ca^{2+}$ ions through the selectivity filter of mutant DHPR(N617D) result in a significantly (p<0.001), ~8 times higher energy barrier compared to wt DHPR (*Figure 6d*), which is in accordance with the experimentally observed complete occlusion of the DHPR(N617D) pore in the presence of physiological concentrations of extracellular $Ca^{2+}$ ions (*Figure 3*). This rather static condition in the DHPR(N617D) pore is well expressed in its lower Hill slope compared to wt DHPR (see *Figure 3d*). The Hill slope/Hill coefficient ($n_H$) derived from four parameter logistic fit of dose-response curve is best portrayed as an 'interaction' coefficient, reflecting the extent of cooperativity among multiple binding sites (*Prinz, 2010*). The considerably more dynamic $Ca^{2+}$ interactions in the wt DHPR pore with its successive short-lived intermediate high and low binding affinities and repulsion mechanisms are consequently apparent in the higher $n_H$ compared to DHPR(N617D).

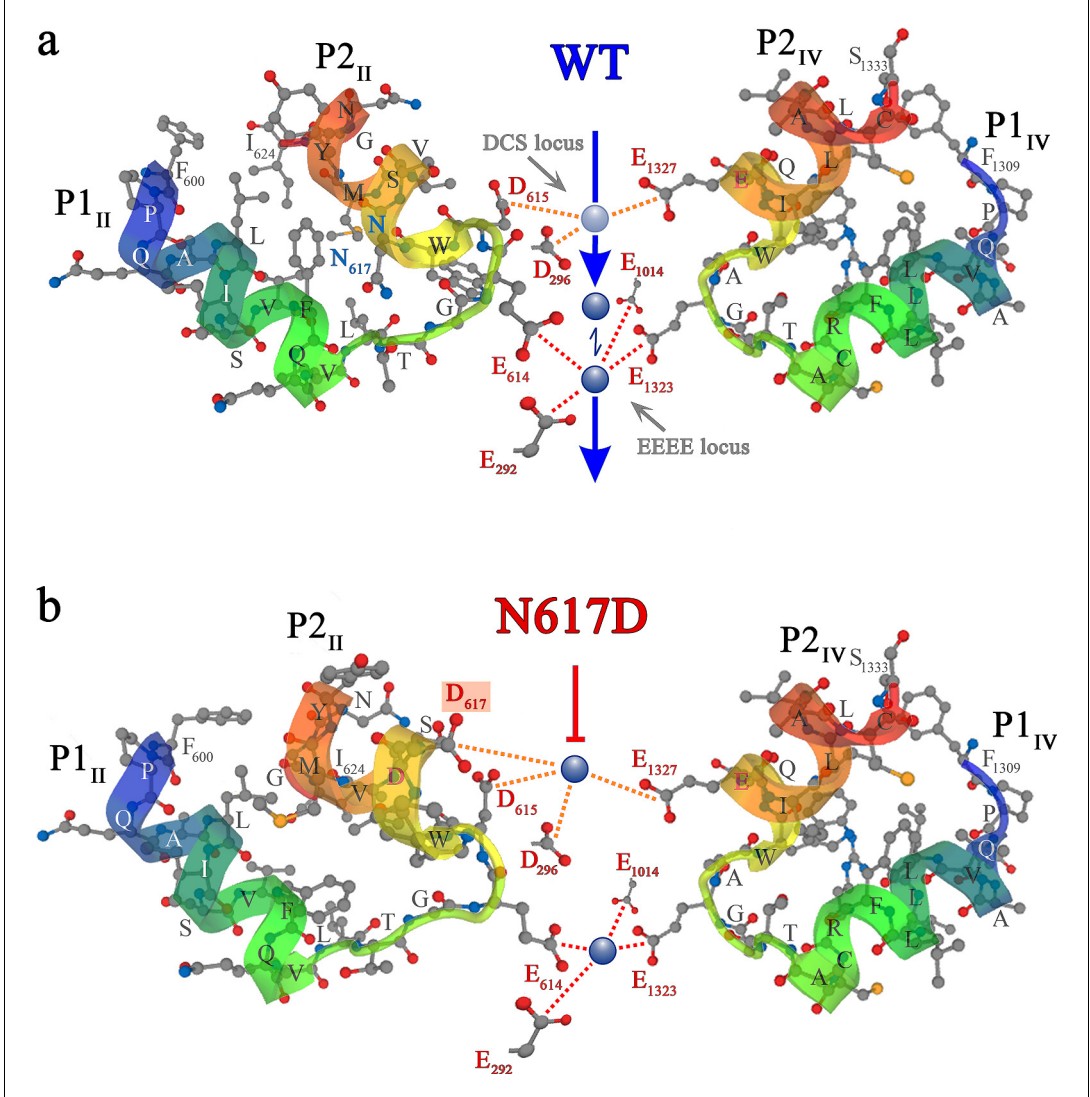

**Figure 5.** Ca²⁺ selectivity and conductance mechanisms in the wt and mutant DHPR(N617D) channel pore. (a, b) De novo conformation prediction of peptide F₆₀₀ - I₆₂₄ constituting the selectivity filter and adjacent pore helices P1 and P2 of DHPRα₁S repeat II (P1_II, P2_II) (*left*) and of peptide F₁₃₀₉ - S₁₃₃₃ forming the opposite repeat IV (P1_IV, P2_IV) (*right*), using the program PEP-FOLD 3.5 (*Thévenet et al., 2012*) on the RPBS web portal. Resulting clusters from 200 independent simulations were sorted by sOPEP energy (*Wang et al., 2011*) to yield the 'best model' prediction. Biasing the model prediction of these peptides by imposing the reference structure of DHPRα₁S according to the Protein Data Bank (PDB accession number 5GJV) (*Wu et al., 2016*) did not lead to major differences compared to unbiased modeling approaches and hence we used unbiased models for the wt (a) and DHPR(N617D) (b) inner channel pore. Depicted best models are graphical overlays of *cartoon* and *balls and sticks* input style options. Models depict the hypothetical mechanism of Ca²⁺ conductance through the wt DHPR (a) and the block of Ca²⁺ conductance due to atypical high Ca²⁺ binding affinity (because of introduction of the negative charge D₆₁₇; b*oxed in red*) in the DHPR(N617D) pore region. *Dotted lines* indicate binding interactions between Ca²⁺ ions (*blue spheres*) and carboxyl oxygens (*red balls*) of glutamate E₂₉₂ and aspartate D₂₉₆ of repeat I, E₆₁₄, D₆₁₅, and D₆₁₇ of repeat II, E₁₀₁₄ of repeat III, as well as E₁₃₂₃ and E₁₃₂₇ of repeat IV. Low affinity Ca²⁺ binding is indicated with a *light blue sphere* and high-affinity binding with *dark blue spheres*. DCS locus is the divalent cation selectivity filter (*Cens et al., 2007*) and EEEE locus is the Ca²⁺ selectivity filter. Vertical *blue arrows* indicate active Ca²⁺ conductance pathway in wt DHPR (a) and *red T-bar* indicates block of Ca²⁺ flux by aberrant high-affinity binding to the DCS locus in the mutant DHPR (N617D) channel pore (b). See *Figure 5—figure supplement 1* for additional blocking strategies of DHPR Ca²⁺ conductance in the evolution of skeletal muscle EC coupling.

The online version of this article includes the following figure supplement(s) for figure 5:

**Figure supplement 1.** Additional blocking strategies of DHPR Ca²⁺conductance in the evolution of skeletal muscle EC coupling.

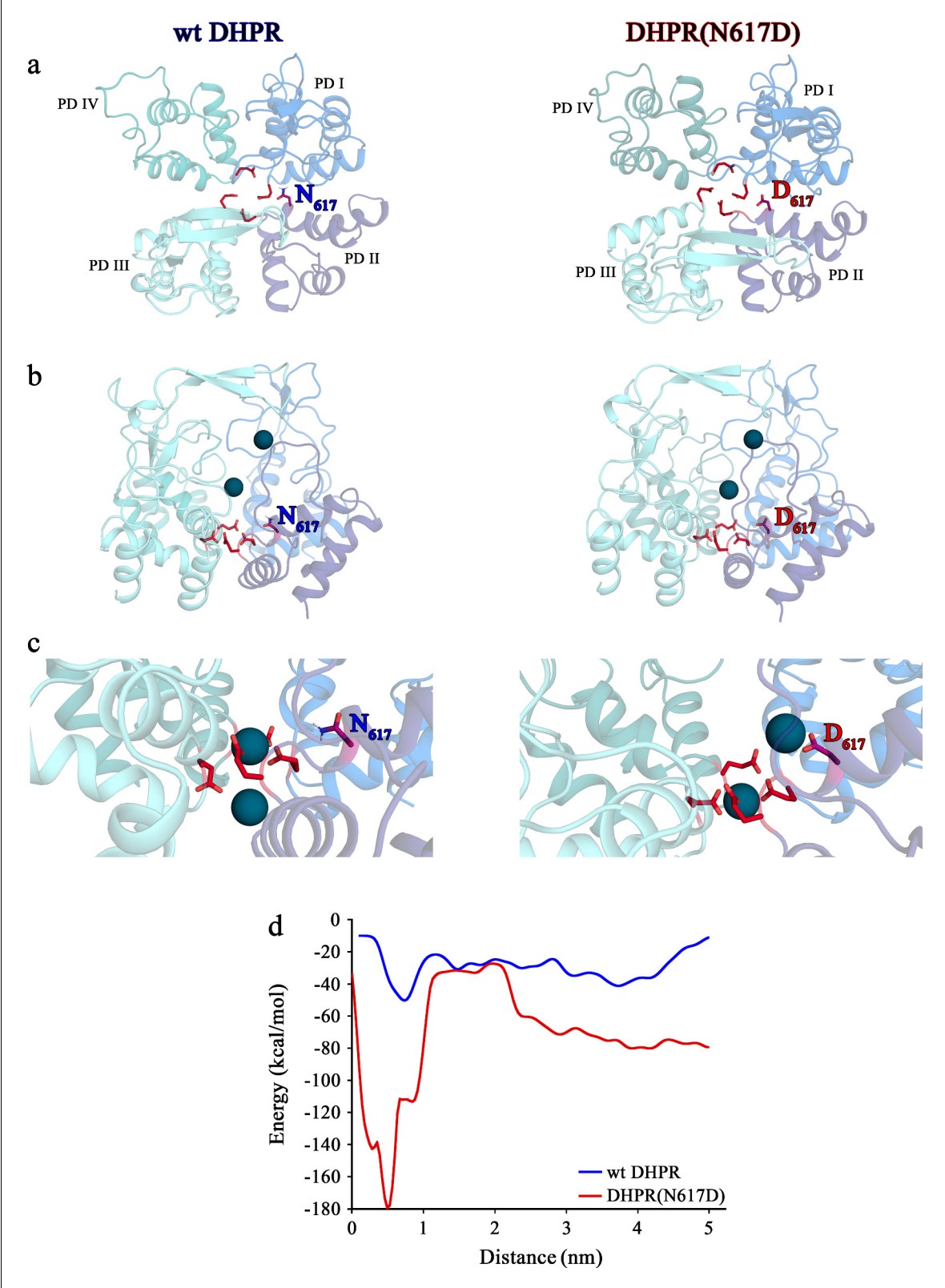

**Figure 6.** Structure models of selectivity filter regions of wt DHPR (*left panels*) and mutant DHPR(N617D) channel pores (*right panels*) showing the movements of $Ca^{2+}$ ions in simulation studies. (a) Top view of the pore illustrating the EEEE and DCS loci. The residues of the EEEE locus are displayed in *red* and the DCS locus is indicated by the position of the residues $N_{617}$ or $D_{617}$. (b) Side view of wt DHPR and mutant DHPR(N617D) pores with $Ca^{2+}$ ions present in the pore before starting the equilibration. The *dark blue spheres* represent van der Waals radii of the $Ca^{2+}$ ions. (c) Snap-
*Figure 6 continued on next page*

*Figure 6 continued*

shots immediately following the equilibration run show that $Ca^{2+}$ ions already moved towards the DCS and EEEE loci. While the front $Ca^{2+}$ ion already leaves the selectivity filter of the wt DHPR toward the cytosolic side, $Ca^{2+}$ ions in the DHPR(N617D) pore are still bound to the DCS and EEEE loci. (**d**) Free energy estimations from metadynamics simulations capturing the movements of $Ca^{2+}$ ions through the selectivity filter region. The free energy profile for the passage of $Ca^{2+}$ ions through wt DHPR selectivity filter is depicted in *blue* and for mutant DHPR(N617D) in *red*. The energy barrier of the $Ca^{2+}$ ion leaving the wt DHPR selectivity filter (15 ± 4 kcal/mol; $n$ = 5) is significantly smaller (p<0.001) compared to DHPR(N617D) (122 ± 20 kcal/mol; $n$ = 5). The process was described by a one-dimensional collective variable that is, the displacement of a $Ca^{2+}$ ion along the axis of the channel pore. A second $Ca^{2+}$ was directly present in the simulation domain. Thus, the energy profile corresponds to the energy experienced by the first $Ca^{2+}$ ion in the presence of the second one. See *Figure 6—video 1* and *Figure 6—video 2* for illustration of the movement of $Ca^{2+}$ ions through the selectivity filter region of wt DHPR and DHPR(N617D) channel pores, respectively.

The online version of this article includes the following video(s) for figure 6:

**Figure 6—video 1.** Movement of $Ca^{2+}$ ions through the EEEE locus of the wt DHPR pore toward the cytosolic side.

**Figure 6—video 2.** Movement of $Ca^{2+}$ ions through the DCS and EEEE loci of the DHPR(N617D) pore.

## Emergence of $Ca^{2+}$ non-permeant DHPRs during evolution

Point mutation N617D implemented for the creation of mouse model *nc*DHPR (*Dayal et al., 2017*) was originally identified to be responsible for DHPR $Ca^{2+}$ non-conductivity in zebrafish fast (glycolytic/white) skeletal muscle (*Schredelseker et al., 2010*). Additionally, with studies on the low-$Ca^{2+}$ conducting DHPR of sterlet (*Acipenser ruthenus*), which is phylogenetically somewhere in between mouse and zebrafish, we showed (*Schrötter et al., 2017*) that during vertebrate evolution (i.e. from the mammalian species, e.g. mouse, to the teleost fishes, e.g. zebrafish) a steady loss of DHPR $Ca^{2+}$ conductance occurred. Subsuming results of several studies, we proposed the hypothesis that during evolution from mammals to teleost fishes an accumulation of DHPR amino acid exchanges occurred that contributed to the reduction of $Ca^{2+}$ conductance (*Schredelseker et al., 2010*; *Dayal et al., 2017*; *Schrötter et al., 2017*). Mutation N→D (N617D; mouse numbering) that finally 'turned off' the already reduced $Ca^{2+}$ conductance evolved only in quite a late phylogenetic stage (*Dayal et al., 2017*; *Schrötter et al., 2017*), following the teleost-specific third round (Ts3R) of gene duplication (*Meyer and Van de Peer, 2005*; *Glasauer and Neuhauss, 2014*). Beside DHPR non-conductivity, the evolutionary pressure that caused additional substantial modifications in skeletal muscle organization and physiology in teleost fishes (*Schredelseker et al., 2010*; *Dayal et al., 2017*; *Schrötter et al., 2017*) arose from the critical demand for tighter controlled, faster and stronger muscle contractions, crucial for high-speed movements in the aquatic prey-predator environment (*Dayal et al., 2019*).

Interestingly, Ts3R headed into the evolution of a second DHPR isoform in zebrafish slow (oxidative/red) skeletal muscle that is likewise $Ca^{2+}$ non-conducting (*Schredelseker et al., 2010*). This slow muscle DHPR is so far the only described innate DHPR with a distorted EEEE locus, where glutamate of repeat I is substituted by glutamine. Exchange of this selectivity filter $E_{292}$ with Q in a GFP-tagged rabbit $DHPR\alpha_{1S}$ clone (*Grabner et al., 1998*) yielded mutant DHPR(E292Q), which upon heterologous expression in *dysgenic* myotubes confirmed the abolishment of inward $Ca^{2+}$ currents (*Schredelseker et al., 2010*) with a slight outward $Cs^+$ current, typically starting at +20 to+30 mV (*Bannister and Beam, 2011*). As described earlier (*Yang et al., 1993*), the EQ pore mutation in repeat I of the cardiac DHPR exerted a minor effect, as the increase in $IC_{50}$ was only twofold compared to the wt. If we assume that a similar right-shift of affinity also holds true for the skeletal muscle DHPR(E292Q), then appropriate $Ca^{2+}$ pore-affinity essential for proper $Ca^{2+}$ selectivity and $Ca^{2+}$ conductance must exist in a surprisingly small range. Incorporation of our present and previously published data (*Yang et al., 1993*; *Schredelseker et al., 2010*) indicates that this small range might be within approximately one order of magnitude, somewhere between 0.37 ($IC_{50}$ for N617D) and 3.2 µM (2-fold $IC_{50}$ for wt). The hampered $Ca^{2+}$ selectivity and conductance mechanism of mutant DHPR(E292Q) (*Figure 5—figure supplement 1a*) is expected to be essentially the same as discussed above for DHPR(E1014K). In brief, low-affinity $Ca^{2+}$ binding to the QEEE locus (*Figure 5—figure supplement 1a*) cannot support the crucial $Ca^{2+}$ - $Ca^{2+}$ repulsion mechanism and thus, $Ca^{2+}$ conductance through mutant DHPR(E292Q) is blocked. Likewise, $Ca^{2+}$ block of the bidirectional flux of monovalent cations, and hence $Ca^{2+}$ selectivity is abolished.

Lastly, a third evolutionary concept also yielding a $Ca^{2+}$ non-conducting DHPR was identified in the fast skeletal muscle of teleost fishes (*Schredelseker et al., 2010*). Although in the early phylogenetic teleost species (including zebrafish from the order *cypriniformes*) mutation N→D (N617D, mouse numbering) is the archetypical mutation to block DHPR $Ca^{2+}$ influx, in phylogenetically higher developed teleost species starting with the order *lophiiformes* (anglerfishes), this negatively charged D was lost by mutating to a neutral T (*Schredelseker et al., 2010*). Concurrent to this D→T mutation, DHPR $Ca^{2+}$ non-conductivity was re-installed by mutation of another D, which is one of the negative charges in the DCS locus (located in pore repeat I) and highly homologous in all mammalian L-type $Ca^{2+}$ channels, to positively charged K (D→K). As demonstrated previously (*Schredelseker et al., 2010*), exchange of this DCS locus $D_{296}$ with K in a GFP-tagged rabbit DHPRα$_{1S}$ clone yielded mutant DHPR(D296K). Upon heterologous expression in *dysgenic* myotubes, this single charge conversion was sufficient to abolish inward $Ca^{2+}$ currents. According to our combined model of $Ca^{2+}$ selectivity and conductance and illustrated in *Figure 5—figure supplement 1b*, $K_{296}$ does not permit formation of an active DCS locus, and thus $Ca^{2+}$ from the t-tubular (extracellular) space is no more attracted to the DCS locus. Resultantly, there is lack of easy to mobilize low-affine DCS-bound $Ca^{2+}$ that would compete with the tightly EEEE-bound $Ca^{2+}$ for the binding valences of the EEEE locus (*Figure 5—figure supplement 1b*). Thus, the $Ca^{2+}$- $Ca^{2+}$ repulsion mechanism (*Sather and McCleskey, 2003*) and pushing out of the $Ca^{2+}$ bound to the selectivity filter into the cytosol cannot take place. The surprising implication of charge conversion D296K in blocking of inward DHPR $Ca^{2+}$ flux proves the importance of the DCS locus for proper inward DHPR $Ca^{2+}$ currents in skeletal muscle and consequently, fundamentally supports our model of DHPR $Ca^{2+}$ selectivity and $Ca^{2+}$ conductivity.

# Materials and methods

## Key resources table

| Reagent type (species) or resource | Designation | Source or reference | Identifiers | Additional information |
|---|---|---|---|---|
| Strain, strain background (*Mus musculus*) | *nc*DHPR | doi:10.1038/s41467-017-00629-x *Dayal et al., 2017* | | |
| Chemical compound, drug | (±)Bay K 8644 | Sigma-Aldrich | Cat#: B112 | 10 µM |
| Chemical compound, drug | Nifedipine | Sigma-Aldrich | Cat#: N7634 | 10 µM |
| Chemical compound, drug | Tetraethylammonium chloride (TEA-Cl) | Sigma-Aldrich | Cat#: T2265 | 145 mM |
| Chemical compound, drug | N-benzyl-p-toluene sulphonamide (BTS) | Santa Cruz Biotechnology, Inc | Cat#: sc-202087 | 100 µM |
| Software, algorithm | MaxChelator simulation program | https://somapp.ucdmc.ucdavis.edu/pharmacology/bers/maxchelator/ | RRID:SCR_018807 | |
| Software, algorithm | ClampFit | Axon Instruments | | version 10.7 |
| Software, algorithm | SigmaPlot | Systat Software, Inc. | RRID:SCR_010285 | version 11.0 |
| Software, algorithm | GraphPad Prism | GraphPad Software, LLC | RRID:SCR_002798 | version 8 |
| Software, algorithm | PEP-FOLD 3.5 | RPBS web portal | | Version 3.5 |
| Software, algorithm | GROMACS | University of Stockholm, University of Upsala | RRID:SCR_014565 | version 2019.2 |
| Software, algorithm | MOE | Chemical Computing Group ULC | RRID:SCR_014882 | version 2020.01 |
| Software, algorithm | AMBER | University of California, San Francisco. | RRID:SCR_014230 | Version 2020 |
| Software, algorithm | PyMOL | Schrödinger, LLC | RRID:SCR_000305 | Version 2.4.0 |

## Animals

Generation of the $Ca^{2+}$ non-conducting (*nc*)DHPR knock-in mouse strain, carrying a point mutation in the *Cacna1s* gene coding for N617D in pore loop II was described previously (*Dayal et al., 2017*). Animal breeding, care and maintenance was conducted in compliance with the guidelines of the EU Directive 2010/63/EU and approved by the Austrian Ministry of Science (BMWF-5.031/0001-II/3b/ 2012). Mice were housed in a controlled environment with a 12/12 hr light/dark cycle and had access to food and water ad libitum.

## Isolation and culture of skeletal myotubes

Primary myoblasts from new born up to 4-day-old pups homozygous for the non-conducting L-type $Ca^{2+}$ channel mutant DHPR(N617D) or wild-type channel were enzymatically isolated and cultured in a humidified 37°C incubator with 5% $CO_2$ as described previously (*Dayal et al., 2017*). Myotubes were maintained in growth medium consisting of Dulbecco's modified Eagle's medium supplemented with 10% fetal calf serum, 10% horse serum, 25 mM HEPES, 4 mM L-glutamine, and 1x penicillin/streptomycin and later replaced with differentiation medium (no fetal calf serum and only 2% horse serum).

## Whole cell patch clamp

Ionic currents were evoked by a standard 200 ms voltage-step protocol from −50 to +80 mV in 10 mV increments from a holding potential of −80 mV (*Dayal et al., 2017*), unless otherwise stated. To reduce inward currents via endogenous $Na_V$ and T-type $Ca^{2+}$ channels, every test pulse was preceded by a 1 s prepulse to −30 mV followed by a 50 ms repolarization to −50 mV (*Adams et al., 1990*). Borosilicate glass patch pipettes had resistance of 2–3 MΩ when filled with (in mM) 145 Cs-aspartate, 2 $MgCl_2$, 10 HEPES, 0.1 $Cs_2$-EGTA, and 2 Mg-ATP (pH 7.4 with CsOH). The standard bath solution for recording $Ca^{2+}$ currents contained (in mM): 10 $CaCl_2$, 145 TEA-Cl and 10 HEPES (pH 7.4 with TEA-OH). Myosin-II blocker BTS (100 µM, Sigma) was constantly present in the bath solution.

To test if depolarization-induced potentiation protocols known to promote mode 2 gating in L-type $Ca^{2+}$ channels could evoke currents through DHPR(N617D), strong or long depolarizations in the presence of racemic 1,4-dihydropyridine (DHP) agonist (±)Bay K 8644 (10 µM) were performed (*Bannister and Beam, 2011*; *Bannister and Beam, 2013*). Pulse protocol for strong depolarization is depicted in *Figure 1b*. Briefly, 200 ms depolarization to either +90 mV or +60 mV is followed by a +60 mV pulse for 100 ms and finally by a repolarization to −20 mV for 70 ms. For long depolarization (*Figure 1c*), prolonged 2 s pulses from +10 - +80 mV in 10 mV increments were applied starting from a holding potential of −80 mV with an intermediate repolarizing step to −50 mV.

To investigate if DHPR(N617D) conducts slow-activating, non-inactivation inward $Na^+$ currents, 145 mM TEA-Cl in standard bath solution was replaced by 145 mM NaCl (pH 7.4 with NaOH) to achieve near physiological $Na^+$ concentration (150 mM). Furthermore, to test if these $Na^+$ currents were also subject to block by $Ca^{2+}$, 10 mM $Ca^{2+}$ was reduced to near physiological 1 mM $Ca^{2+}$.

To assess $Ca^{2+}$ pore-binding affinity, dose-inhibition experiments for $Ca^{2+}$ block of inward $Li^+$ currents were performed. The bath solution for recording $Li^+$ currents contained (in mM): 100 LiCl, 10 HEPES, 10 EGTA, and 25 for $CaCl_2$ plus TEA-Cl (pH 7.4 with TEA-OH). Desired free $Ca^{2+}$ concentrations (0 to 30 µM) were obtained by calibrating $CaCl_2$ and TEA-Cl concentrations calculated using the MaxChelator simulation program (https://somapp.ucdmc.ucdavis.edu/pharmacology/bers/max-chelator/) (Supplementary File *Table 1*).

To test if the inward $Li^+$ currents under external free $[Ca^{2+}]=0$ as well as the slow outward and fast inward currents recorded under external 150 mM $Na^+$ and 1 mM $Ca^{2+}$ are mediated by DHPR (N617D) in *nc*DHPR myotubes, 10 µM of the 1,4-DHP antagonist nifedipine was added to the respective bath solutions.

All recordings were performed at room temperature using the Axopatch 200B amplifier (Axon Instruments Inc, CA), filtered at 1 kHz and sampled at 5 kHz.

## Data and statistical analysis

Data were analyzed and plotted using ClampFit (v10.7; Axon Instruments), SigmaPlot (v11.0; Systat Software, Inc) and Prism 8 (GraphPad Software, LLC). Data are represented as mean ± SEM and

$n$ = number of myotubes. Statistical significance was calculated using unpaired Student's $t$-test, unless otherwise stated and was set as follows: *$p<0.05$, **$p<0.01$, and ***$p<0.001$.

## Structure preparation and molecular dynamics simulations

Atomic models were based on the cryo-EM structure of the rabbit DHPRα$_{1S}$ - verapamil complex with a dilated intracellular gate associated to the binding of the phenylalkylamine $Ca^{2+}$ antagonist drug verapamil (PDB accession number 6JPA) (*Zhao et al., 2019*). The structure of mutant DHPR (N617D) was derived from wt DHPR structure by replacing N$_{617}$ with the negatively charged residue D$_{617}$ and carrying out a local energy minimization using MOE (Molecular Operating Environment, Chemical Computing Group, version 2020.01). For simulations, we removed the voltage-sensing domains and truncated the S5 and S6 helices of each repeat, keeping the last nine residues of the S5 and S6 helices. The C- and N-termini of each repeat were capped with acetylamide (ACE) and N-methylamide to avoid perturbations by free charged functional groups. The starting structures for simulations were prepared in MOE using the Protonate3D tool (*Labute, 2009*). To neutralize the charges, we used the uniform background charge (*Case et al., 2020*; *Hub et al., 2014*). Using the tleap tool of the AmberTools20 package (*Case et al., 2020*; *Roe and Cheatham, 2013*), crystal structures were soaked in cubic water boxes of TIP3P water molecules with a minimum wall distance of 10 Å to the protein (*Jorgensen et al., 1983*; *El Hage et al., 2018*; *Gapsys and de Groot, 2019*). We added a total of 10 $Ca^{2+}$ ions, corresponding to a concentration of approximately 10 nM. For all simulations, parameters of the AMBER force field 14 SB were used (*Maier et al., 2015*). The structures were carefully equilibrated using a multistep equilibration protocol (*Wallnoefer et al., 2011*).

For both wt DHPR and mutant DHPR(N617D), 10 ns of molecular dynamics (MD) simulations were performed in an isothermal - isobaric (NpT) ensemble using the GPU MD simulation engine pmemd. cuda (*Salomon-Ferrer et al., 2013*) to further equilibrate the structures in the presence of the $Ca^{2+}$ ions. Bonds involving hydrogen atoms were restrained by applying the SHAKE algorithm (*Miyamoto and Kollman, 1992*), allowing a time step of 2 fs. Atmospheric pressure of the system was preserved by weak coupling to an external bath using the Berendsen algorithm (*Berendsen et al., 1984*). The Langevin thermostat (*Doll et al., 1975*; *Adelman, 1976*) was used to maintain the temperature at 300 K during simulations.

## Metadynamics simulations

Metadynamics is a powerful method to explore the properties of multidimensional free energy landscapes and to enhance the sampling of configurational space in reasonable computing time (*Barducci et al., 2011*). Metadynamics reconstructs the free energy surface as a function of few selected degrees of freedom, referred to as collective variables (CV), which accelerate rare events in the systems. The CVs should be able to characterize the key features of physical behavior of interest, distinguish between all different metastable states, and include the slow degrees of freedom. In metadynamics, an external history-dependent repulsive bias potential function constructed as a sum of Gaussians is deposited along the trajectory in the CV space and thereby, discourages revisiting and oversampling of same configurations. For metadynamics simulations, we used the GROMACS version 2019.2. The aim of the metadynamics simulation was to capture the movement of $Ca^{2+}$ ions along the selectivity-filter conducting pathway and their passing through the EEEE motif. As CV, we chose the distance between the center of masses (COM) of the EEEE motif residues and the upper $Ca^{2+}$ ion.

Simulations were performed at 300 K in an NpT ensemble. We used a Gaussian height of 1.5 kJ/mol and width of 0.1 nm. For both wt DHPR and mutant DHPR(N617D), five repetitions of metadynamics runs, each 10 ns, were performed. Pymol Molecular Graphics System was used to visualize the key interactions and differences between wt DHPR and mutant DHPR(N617D) pore conductances.

## Acknowledgements

This study was supported by the Austrian Science Fund (Fonds zur Förderung der Wissenschaftlichen Forschung, FWF) Research grants P23229-B09 (to M.G.), P27392-B21 (to MG and AD).

## Additional information

### Funding

| Funder | Grant reference number | Author |
|---|---|---|
| Austrian Science Fund | P23229-B09 | Manfred Grabner |
| Austrian Science Fund | P27392-B21 | Anamika Dayal<br>Manfred Grabner |

The funders had no role in study design, data collection and interpretation, or the decision to submit the work for publication.

### Author contributions

Anamika Dayal, Manfred Grabner, Conceptualization, Resources, Data curation, Software, Formal analysis, Supervision, Funding acquisition, Validation, Investigation, Visualization, Methodology, Writing - original draft, Project administration, Writing - review and editing; Monica L Fernández-Quintero, Data curation, Formal analysis, described the simulation experiments; Klaus R Liedl, Supervision and description of the simulation experiments

### Author ORCIDs

Anamika Dayal (iD) https://orcid.org/0000-0001-8075-8812
Monica L Fernández-Quintero (iD) http://orcid.org/0000-0002-6811-6283
Klaus R Liedl (iD) http://orcid.org/0000-0002-0985-2299
Manfred Grabner (iD) https://orcid.org/0000-0002-5196-4024

### Decision letter and Author response
Decision letter https://doi.org/10.7554/eLife.63435.sa1
Author response https://doi.org/10.7554/eLife.63435.sa2

## Additional files

### Supplementary files

• Supplementary file 1. Composition of external solutions with different free $Ca^{2+}$ concentrations used for recording $Ca^{2+}$ block of inward $I_{Li+}$. The indicated free $Ca^{2+}$ concentrations in the bath solutions were achieved by adjusting the concentrations of $CaCl_2$ and TEA-Cl, calculated using the Max-Chelator simulation program (https://somapp.ucdmc.ucdavis.edu/pharmacology/bers/maxchelator/).

• Transparent reporting form

### Data availability

All data generated or analyzed during this study are included in the manuscript and supporting files.

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
