## [Decision Letter]

**Acceptance summary:**

This study provides convincing experimental evidence for the biophysical mechanism responsible for lack of Ca^2+^ conductance of the zebrafish skeletal muscle dihydropyridine receptor (DHPR). The authors show that introduction of the residue responsible for eliminating Ca^2+^ conduction of the zebrafish DHPR (N617D) into the mouse skeletal muscle DHPR abolishes Ca^2+^ and monovalent cation conduction by causing a 4.2-fold increase in high affinity Ca^2+^ binding and block of the channel pore. This mechanism of inhibition of Ca^2+^ permeation is distinct from that achieved by a mutation (E1014K) within the selectivity filter, in which the channel is impermeable to Ca^2+^ but still conducts monovalent cations. The results provide insight into both how ion conduction is eliminated in the DHPR of teleost fish and the surprisingly different phenotypes of two knock-in mouse models (N617D and E1014K) that abolish DHPR Ca^2+^ influx based on these two distinct mechanisms.

**Decision letter after peer review:**

Thank you for submitting your article "Pore mutation N617D in the skeletal muscle DHPR blocks Ca^2+^ influx due to atypical high-affinity Ca^2+^ binding" for consideration by *eLife*. Your article has been reviewed by 3 peer reviewers, including Henry M Colecraft as the Reviewing Editor and Reviewer #1, and the evaluation has been overseen by Kenton Swartz as the Senior Editor. The following individual involved in review of your submission has agreed to reveal their identity: Stephen Cannon (Reviewer #2).

The reviewers have discussed the reviews with one another and the Reviewing Editor has drafted this decision to help you prepare a revised submission.

Summary:

The manuscript identifies a novel mechanism for abolishing Ca permeation through voltage-dependent Ca (CaV) channels. The authors show that the N617D mutation in CaV1.1 introduces a second high-affinity Ca binding site (Kd ~0.4 uM) at a divalent cation selectivity (DCS) that is separate from the high affinity site (EEEE) within selectivity filter (kd ~1.6 uM), and that the two sites do not exhibit significant electrostatic repulsion when Ca is bound. The results show that skeletal myotubes from N617D knock-in mice do not exhibit detectable inward Ca^2+^ currents even under conditions known to augment Ca^2+^ conduction of WT DHPR channels (DHP agonist, prolonged/sustained depolarization) and also do not exhibit detectable inward Na^+^ conductance in the presence of a physiological Na^+^ concentration even when extracellular Ca^2+^ is reduced to 1 mM. The mechanism of inhibition of Ca^2+^ permeation is distinct from that achieved by mutation (E1014K) within the selectivity filter. The results provide insight into both how ion conduction is eliminated in the DHPR of teleost fish and the surprisingly different phenotypes of two knock-in mouse models (N617D and E1014K) that abolish DHPR Ca^2+^ influx based on these two distinct mechanisms.

Essential revisions:

Two main areas were identified as points needed to be addressed by the authors to strengthen the conclusions of the work.

1. The conclusions that the inward/outward currents observed in Figure 2 are not mediated by N617D channels, while the inward Li+ currents in the absence of Ca^2+^ in Figure 3 are mediated by N617D channels should be directly verified with the use of a DHP antagonist.

2. The final section of the paper presents a conceptual model for how the apparent increase in Ca binding affinity severely reduces Ca flux (i.e. "block") and at the same time Ca occupancy of the EEEE motif preserves selectivity over monovalent cations. The key concept is that N617D enhances the Ca binding at the DCS site (0.4 μm affinity) which lies in the pore vestibule external to EEEE. It is proposed that this high-affinity binding at DCS impedes Ca transition to the EEEE site. A potential problem with this model is that if the DCS → EEEE translocation of Ca is too slow to allow "double occupancy" of EEEE by Ca (and thereby reduce EEEE affinity to mM range and promote flux) then how does the EEEE site get occupied with single Ca? The on-rate for initial block of Li current when adding Ca would be very slow. Quantitative simulation would be very helpful to demonstrate it is possible for the DCS → EEEE translocation of Ca to be slow enough to prevent a detectable Ca conductance, and yet at the same time be sufficiently fast to establish Ca block of Li current in a reasonable amount of time.

---

## [Author Response]

Essential revisions:Two main areas were identified as points needed to be addressed by the authors to strengthen the conclusions of the work.1. The conclusions that the inward/outward currents observed in Figure 2 are not mediated by N617D channels, while the inward Li+ currents in the absence of Ca^2+^ in Figure 3 are mediated by N617D channels should be directly verified with the use of a DHP antagonist.

In agreement with the reviewers’ request, we performed additional patch-clamp experiments in the presence of 10 µM of the 1,4-DHP antagonist nifedipine in the external recording solution to directly verify that the inward / outward currents in Figure 2 are not mediated by DHPR(N617D) channels, whereas the inward Li^+^ currents in the absence of Ca^2+^ in Figure 3 are mediated by DHPR(N617D). Both our claims could be verified by these experiments and according text blocks have been added to the Results (lines 189-198; 226-231; 235-237) and Material and Methods (lines 490-493) sections. Additionally, a new figure panel (Figure 2c) and a new figure (Figure 4) with the corresponding legends have been included.

2. The final section of the paper presents a conceptual model for how the apparent increase in Ca binding affinity severely reduces Ca flux (i.e. "block") and at the same time Ca occupancy of the EEEE motif preserves selectivity over monovalent cations. The key concept is that N617D enhances the Ca binding at the DCS site (0.4 μm affinity) which lies in the pore vestibule external to EEEE. It is proposed that this high-affinity binding at DCS impedes Ca transition to the EEEE site. A potential problem with this model is that if the DCS → EEEE translocation of Ca is too slow to allow "double occupancy" of EEEE by Ca (and thereby reduce EEEE affinity to mM range and promote flux) then how does the EEEE site get occupied with single Ca? The on-rate for initial block of Li current when adding Ca would be very slow. Quantitative simulation would be very helpful to demonstrate it is possible for the DCS → EEEE translocation of Ca to be slow enough to prevent a detectable Ca conductance, and yet at the same time be sufficiently fast to establish Ca block of Li current in a reasonable amount of time.

In order to address these queries, we performed molecular dynamics (MD) simulations in combination with an enhanced sampling technique (described in the Materials and methods section of the revised manuscript; lines 501-542) to estimate the differences in free energy profiles of Ca^2+^ ions passing through the selectivity filters of wt DHPR and mutant DHPR(N617D). We found that in wt DHPR immediately after the equilibration phase, two Ca^2+^ ions compete for the EEEE locus. Consequently, Ca^2+ –^ Ca^2+^ repulsion occurs in this doubly occupied EEEE locus, so that the first of the two Ca^2+^ ions is pushed towards the cytosolic side. Moreover, metadynamics simulations capturing the passage of Ca^2+^ ions through the pore revealed a significantly smaller energy barrier for the wt DHPR selectivity filter compared to mutant DHPR(N617D).

MD simulations on mutant DHPR(N617D) revealed a different picture. Already 1 ns after the equilibration phase, we find one Ca^2+^ ion stabilized in the EEEE locus and the other bound to the DCS locus. This shows that already in a very short equilibration phase, the Ca^2+^ ions reach and occupy the EEEE and DCS loci, and since ions cannot pass though the selectivity filter regions anymore, this result in complete occlusion of the pore. This additional stabilization at the DCS locus that a Ca^2+^ ion experiences while moving towards the cytosolic side is also reflected in the ~ 8-times higher energy barrier compared to wt DHPR. The slow detachment and thus slow migration of the Ca^2+^ ions towards the cytosolic side, as observed in our simulation study, suggests occlusion of the pore without the force applied during simulations to pull the Ca^2+^ ions through the pore.

Observations from the simulation experiments are presented in a new figure (Figure 6, with 2 videos as figure supplement) and the Discussion section (lines 356-365, and 373-383).

As a consequence of these new insights, we removed the sentence “Additionally, undisturbed continuous high affinity binding of Ca^2+^ to the EEEE locus results in complete occlusion of the DHPR(N617D) pore, even for monovalent cations, as demonstrated above” (lines 435-437 of the original manuscript) from the Discussion section. Misleadingly it suggested that divalent and monovalent cations can only be blocked at the continuously Ca^2+^- occupied EEEE locus in mutant DHPR(N617D), a statement for which no experimental evidence exists. Considering the tight dimension of the pore around the DCS locus, it is very likely that cations are blocked already at the Ca^2+^- occupied DCS locus. This mechanism would allow a very fast on-rate for the initial block of Li^+^ currents when adding Ca^2+^ ions to the external solution, independent of the speed of the DCS → EEEE translocation, which can be slow enough to prevent detectable Ca^2+^ conductance. Detailed, quantitative MD simulations with reliable time scales and interaction energies to predict if the DHPR(N617D) channel pore can be blocked via the Ca^2+^- occupied DCS locus is a challenging aim, but considering the massive need of computing resources would require an independent study.